

# A new cooperative control solution of subway BAS: an improved fuzzy PID control algorithm

Hui Fang[1,2], Shusong Yang[1], Ying Shi[3], Yang Wang[1], Yue Jiang[1], Chaochao Song[3] and Wei Zhang[2]

[1] Ningbo Rail Transit Group Co., Ltd., Ningbo, Zhejiang, China
[2] Zhejiang University, Hangzhou, Zhejiang, China
[3] New Business Incubation Department, SUPCON Technology Co., Ltd., Hangzhou, Zhejiang, China

Corresponding authors
Hui Fang, stheseus@126.com
Wei Zhang, zwei9876@163.com

## ABSTRACT

The building automation system (BAS) of a subway is the core component of monitoring and managing urban rail transit systems. For the current problems such as low control efficiency, insufficient accuracy, and poor stability of metro BAS, this article proposes a cooperative control framework based on an improved fuzzy proportional-integral-derivative (PID) algorithm. Firstly, the concept of an integrated supervisory control system (ISCS) for subways is introduced by summarizing the previously implemented engineering construction and combining it with advanced automation technology. The system's overall design under the ISCS framework is also improved by integrating it with the fire alarm system (FAS) with the BAS as the core unit of the reliance. Then, an improved seeker optimization algorithm (ISOA) is employed to optimize the parameters of the fuzzy PID control algorithm to achieve a coordinated control of the system based on considering the time lag problem. Finally, the accuracy, efficiency, and stability of the coordinated control response of the BAS under the ISCS framework are tested experimentally. The results suggest that the proposed cooperative control solution of BAS employing the improved fuzzy PID algorithm has good control accuracy and response efficiency and can also ensure the BAS's higher stability in the coordinated control process, which thus greatly improves the automation level of the subway and provides a safer and more reliable high-performance for the ISCS of the subway in the urban rail transportation industry.

## INTRODUCTION

The construction of subways has stepped into a rapid development phase since its fast and efficient urban public transportation saves time and money for millions. As an important means for each city to resolve the traffic congestion problem led by rapidly expanded urban areas and to boost domestic demand to accelerate economic growth are some leading advantages. The internal system of the subway has also received growing attention (*Yu et al., 2018*; *Zhixin et al., 2010*). The subway is generally built below the ground.

Commuters gather in public spaces whose carrying capacities are relatively limited. Several electromechanical equipment are located in the stations and subway zones. Those facilities help commuters travel safely and comfortably, and traffic flowing in stations continues smoothly due to their operational capability. Many types and numbers of electromechanical equipment with complicated control processes are distributed along the line. The building automation system (BAS) is an automatic control system based on the theory of ventilation, air conditioning, and disaster prevention in the subway system that combines computer and network technology with the principle of automatic control of electromechanical equipment and uses a distributed microcomputer monitoring system. The main functionalities are air conditioning and ventilation, lighting, water supply and drainage, escalators, elevators, and guidance signs in subway stations and tunnels (*Jia et al., 2010*). Thus, passengers travel safely and comfortably while ensuring the safety and security of the stations. However, various scattered devices in the current subway system and the existing BAS often have difficulty responding quickly to the actual changes in the subway stations since low control accuracy and poor stability mainly appear as the main issues. With the increasing demand for metro automation control, the realization of intelligent coordinated control of metro BAS has become a key research direction for both academia and industry (*Guo, Han & Li, 2010*).

In recent years, as computerized systems have advanced, control and network communication technologies have been employed by domestic and foreign metro companies that used a hot standby redundant programmable logic controller (PLC) to build BAS by utilizing metro design specifications (*Basnayake et al., 2017*; *Vallati et al., 2016*; *Ohlenbusch et al., 2018*). The scientific and reasonable integration and interconnection of relevant subsystems in the integrated supervisory control system (ISCS) is essential for the system hardware, software platforms, and network construction (*Rajeh, 2022*; *Ahmadi-Assalemi et al., 2022*; *Hussain et al., 2022*). From the current status of a subway BAS implementation, the structure of a subway ISCS depends on its data service mode since both comparison and analysis of the subway lines in different periods are required. In the past, the centralized ISCS mainly relied on the performance of the central-level server. Then, the central server and the network carried a large burden, which could easily cause the paralysis of the whole line in case major accidents or occurrences happened (*Yan, Yang & Liu, 2022*; *Chand et al., 2022*; *Parveen & Nandan, 2022*). With the rapid development of network technology, a distributed ISCS has been used in large numbers in the current metro industry and has become the current development direction of the urban rail transit industry.

Further study reveals that BAS and the fire alarm system (FAS) are set up independently in a conventional scheme, *i.e.*, the two systems are configured with various devices at the central control level and station level and use independent transmission channels for information sharing. Each system performs its monitoring separately. Hence, an integrated BAS and FAS, which cooperate and coordinate with each other in practice, are characterized as complementary and dependent whereas a single system cannot control the whole process. The ISCS has access to the professional data of the subsystem of the subway station, which can fully combine the information of trains entering and leaving

stations and the pedestrian flow, as well as the information of the subway obtained by BAS sensors, to perform periodic controls of the automatic adjustment and the electromechanical equipment. Thus, the purpose of reducing energy consumption and providing passengers with a high-quality riding environment could be achieved. Therefore, it is of great significance for engineering practices and economic value to study the coordinated control framework of metro BAS when the ISCS is under consideration.

The article is divided into five main parts: the first part is the introduction, the second part is the design of a BAS coordination control framework when the ISCS framework is considered, the third section is the application of the fuzzy proportional-integral-derivative (PID) control algorithm and its optimization in a BAS coordination control framework, the fourth section is the experimental validation, and the fifth part is the conclusion and future work.

## THE DESIGN OF A BAS COORDINATION CONTROL FRAMEWORK UNDER THE ISCS FRAMEWORK

### The general structure of the ISCS

As a comprehensive information platform, ISCS integrates the central-level functions of several subsystems such as signaling and automatic ticketing to master the operation of the whole line equipment and is also responsible for monitoring and dispatching the equipment within its assessment and maintenance personnel. The ISCS can monitor the operation status and fault situations of the equipment under the assessment of each station system of the whole line. Issues are instructed to each station to unify the command and coordinate the operation of each station.

The overall structure of the ISCS is designed based on the principles of two levels of management (central, station) and three levels of control (central, station, site), as shown in Fig. 1.

The ISCS is divided into three layers from the configuration of hardware equipment.
1) Central-level Integrated Supervisory Control System (CISCS).
2) Station-level Integrated Supervisory Control System (SISCS).
3) Site-level control equipment (integrated subsystem part).

### The overview of the coordination control framework

BAS generally consists of the BAS equipment installed in the station's electric control rooms and other related components such as vehicle sections and on-site BAS equipment. BAS equipment is composed of a redundant station controller (PLC controller), BAS bus network, remote IO, and IBP disk PLC.

The PLC near the station control room (end A) is the master controller and the PLC at the other end (end B) is the slave controller. The PLCs at the two ends are connected by redundant buses, and various types of RI/O, site devices with intelligent communication ports, and local site mini-controllers are connected to monitor and manage the electromechanical equipment (heating, ventilation, and air conditioning (HVAC), escalators, low-voltage lighting, water supply, and drainage) at the two ends of the station, respectively.

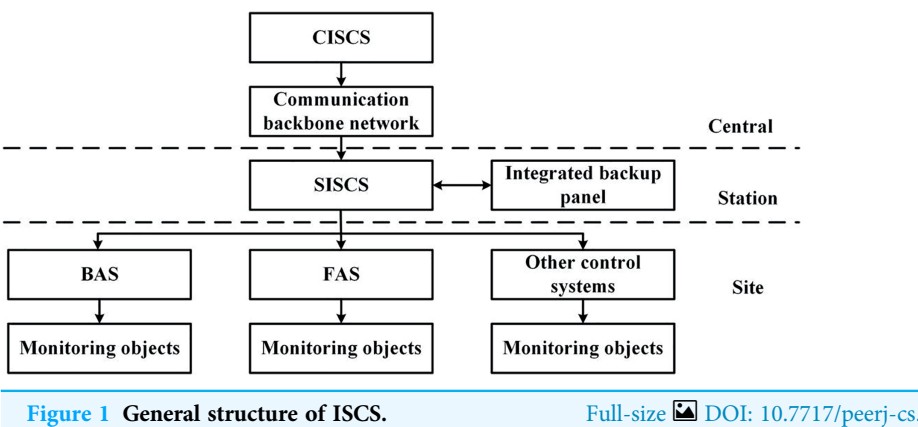

**Figure 1 General structure of ISCS.**

In the fire mode, the FAS sends a command to the BAS, and the BAS controller will start the relevant equipment in a disaster mode according to the predefined working conditions. Then, the station control room has an integrated backup panel (IBP) set up by the integrated monitoring system as an emergency backup control in a disaster mode. The BAS has a single PLC in the IBP panel, which requires high reliability and real-time operation of the BAS and requires the system response faster and more accurate as shown in Fig. 2.

## The architecture of the framework of the coordination control

The BAS adopts a two-level management and three-level control architecture, *i.e.*, two-level management at the control center and station (vehicle section and parking lot), and three-level control at the control center (vehicle section, station, and parking lot). The BAS is integrated into the comprehensive monitoring system at the station level, which realizes its functions at the central and station levels, respectively. The BAS comprises rooms such as control, monitoring equipment, station control, vehicle section, parking lot, and other on-site BAS equipment. The control structure of a BAS is shown in Fig. 3.

Two sets of redundant G5pro PLCs are installed in the loop control rooms at each end of the station, with the PLC near the station control room (end A) as the master controller and the PLC at the other end (B end) as the slave controller. The PLCs at both ends of the station are connected to the Ethernet ring network. All kinds of RI/O and field devices with intelligent communication ports are connected to monitor and manage the electromechanical equipment (ventilation and air conditioning, lighting, directional signs, escalators, elevators, human-proof doors (flood-proof doors) and water supply and drainage systems at both ends of the station. The architecture of the coordinated control framework of a BAS is shown in Fig. 4.

The BAS station is equipped with a special COM5004RTU communication module connected to the FAS station for communication. In the fire mode, the FAS sends a command to the BAS, and the BAS controller will switch to the disaster mode and activate the relevant equipment according to the predetermined working conditions. The station
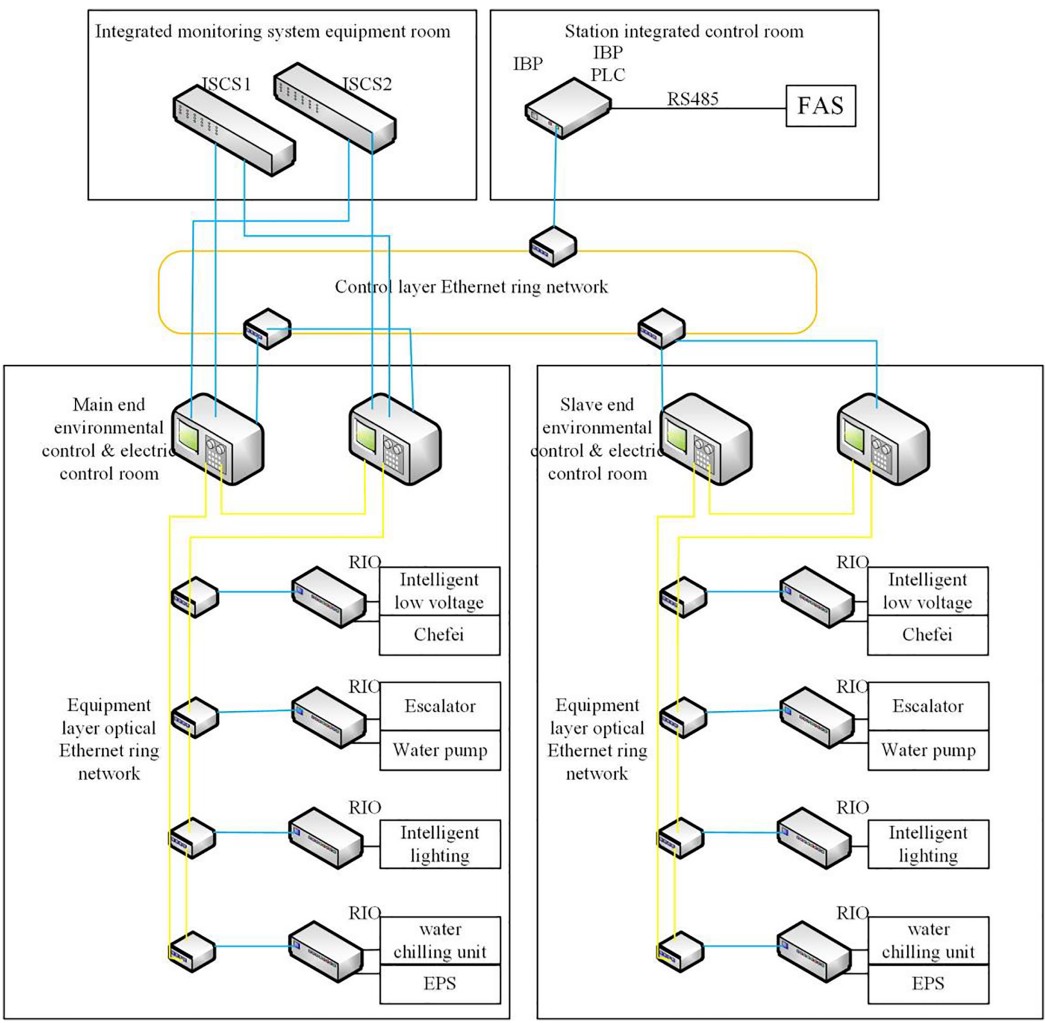

**Figure 2  The actual control structure of ISCS.**

control room has an IBP set up by the integrated monitoring system as an emergency backup control in disaster mode. The BAS site-level monitoring system mainly includes a redundant GCU5001-S controller as the BAS redundant controller, and the site-level monitoring equipment is configured according to the installation of mechanical and electrical equipment. The G5pro series remote intelligent RI/O is equipped to communicate with the redundant GCU5001-S controller through the network.

The main PLC controller of the underground station is equipped with two redundant EtherNet/IP Ethernet communication interfaces (redundant COM5002TCP modules) to communicate with the redundant switches of the integrated monitoring system. The GCU5001-S PLC controllers at both ends of the station communicate with the Ethernet ring network by using fiber optic media. The GCU5001-S PLC controllers communicate with the G5pro series remote I/O modules. GCU5001-S PLC controller and G5pro series remote I/O, frequency converter, intelligent low-voltage, *etc.*, are connected by a ring

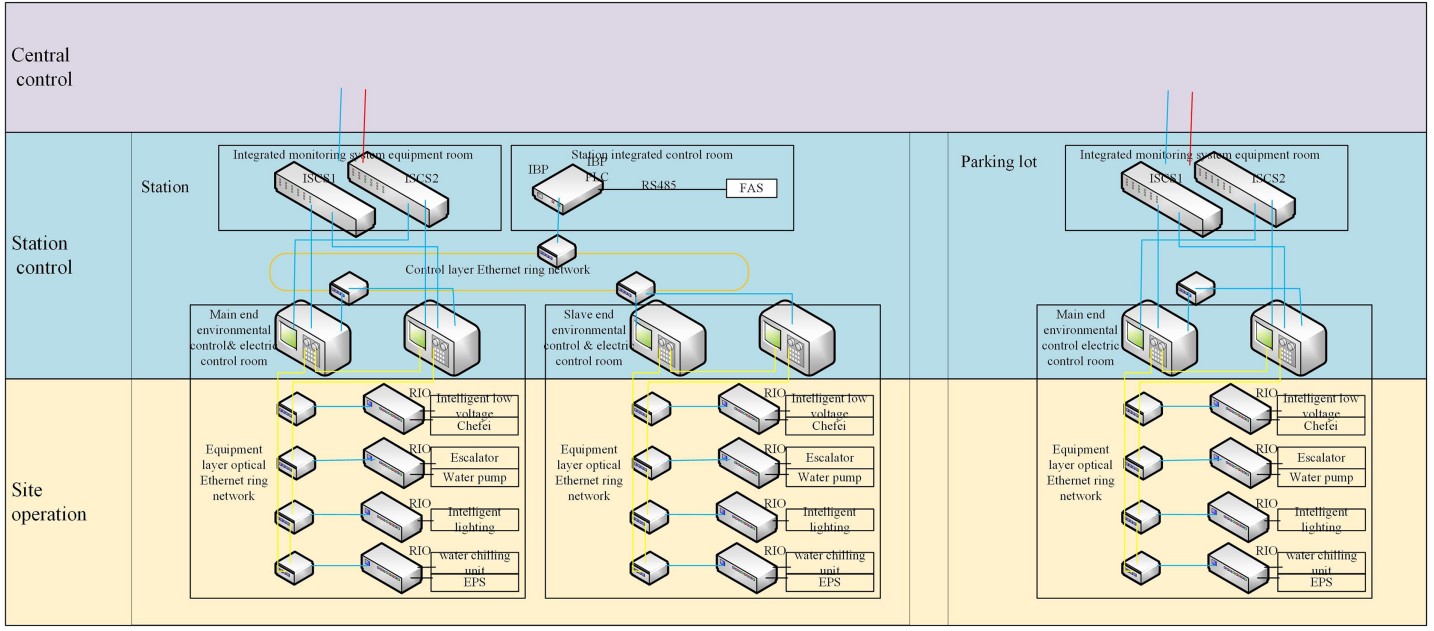

**Figure 3 The control structure of the BAS.**

network. The BAS redundant GCU5001-S PLC controller is connected to the FAS *via* the RS485 bus. The IBP disk GCU5001-S PLC is connected to the master and slave GCU5001-S PLC controllers *via* the Ethernet ring network.

# THE IMPROVED FUZZY PID CONTROL ALGORITHM

## Fuzzy PID control algorithm

Fuzzy control is a nonlinear control method, which has the advantages of not requiring an exact mathematical functional model, high stability, strong robustness, and better control systems containing PLCs. Therefore, to achieve the coordinated control of a BAS, this article adopts the fuzzy PID control algorithm as the basic control algorithm of a BAS.

The system structure of the fuzzy PID controller is shown in Fig. 5. The system mainly consists of two parts: PID control and fuzzy control. $R_p$ denotes the preset coordinated control response effect, and $R_a$ represents the coordinated control response effect in actual action, and the error $\varepsilon$ and the change error rate $\varepsilon_c$ are used as the input signals of the fuzzy controller (*Laib et al., 2022*; *Ahmadi Kamarposhti et al., 2022*).

The PID controller output value is calculated by using Eq. (1) (*Puviyarasi, Murukesh & Srividya, 2022*).

$$u(t) = k_p \left[ e(t) k_i \int_0^t e(t)dt + k_d \frac{d\varepsilon(t)}{dt} \right] \tag{1}$$

where $u(t)$, $k_p$, $k_i$, and $k_d$ denote controller output value, proportionality, integration, and differentiation coefficients, respectively in Eq. (1).

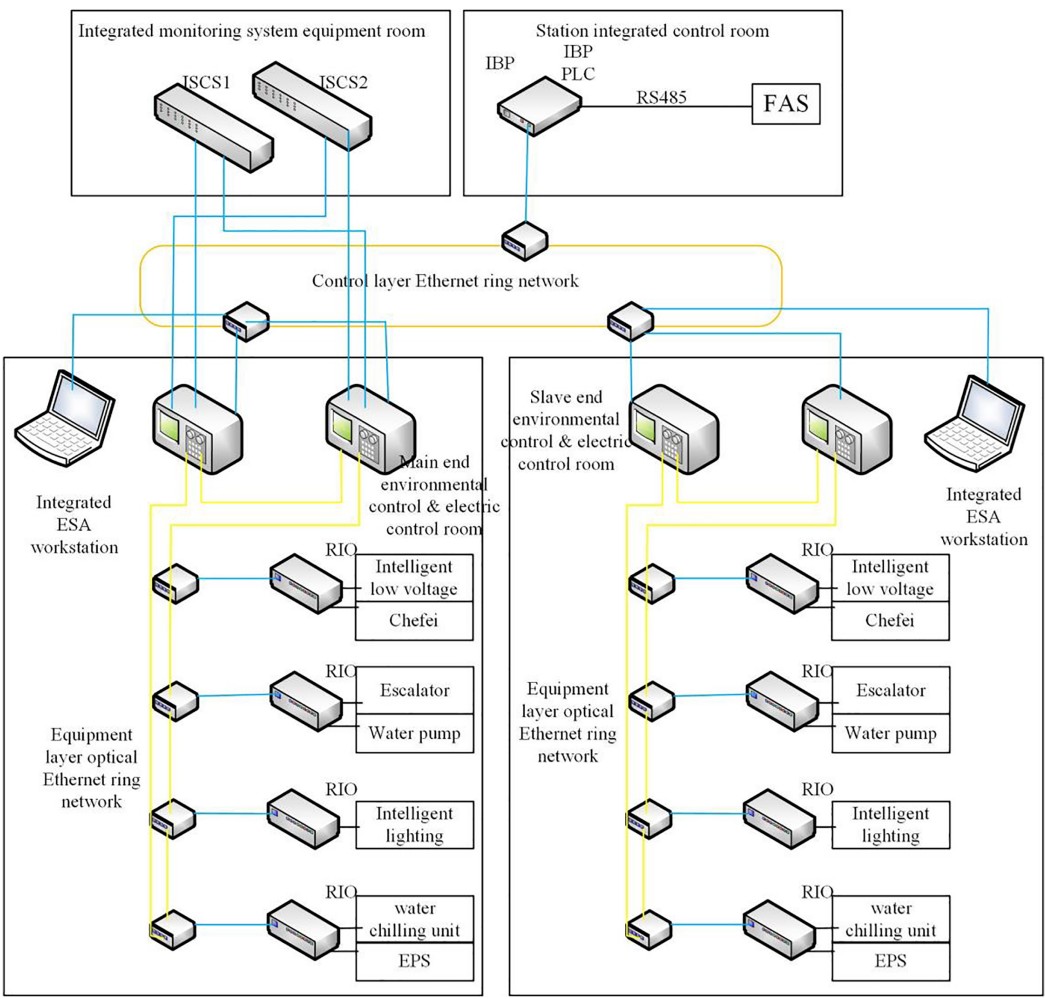

**Figure 4 Coordinated control solution architecture for BAS.**

When the BAS responds to the coordinated control command, it detects the response error $\varepsilon$ and the change error rate $\varepsilon_c$ in real-time and then finds the optimal solution for adjusting the PID control parameters of $\triangle k_p$, $\triangle k_i$, and $\triangle k_d$ according to the fuzzy rules in real-time. The system response is fast and smooth. The adjustment is shown in Eq. (2) (*Wang, Lu & Wang, 2022*).

$$\begin{cases} k_p = k_{p1} + \triangle k_p \\ k_i = \quad k_{i1} + \triangle k_i \\ k_d = k_{d1} + \triangle k_d \end{cases} \qquad (2)$$

In Eq. (2), $k_{p1}$, $k_{i1}$ and $k_{d1}$ are the initial values of the PID control parameters of $k_p$, $k_i$, and $k_d$, respectively. The theoretical domain of the input variables' errors $\varepsilon$ and change error rate $\varepsilon_c$ is contained in [−6, 6], and the theoretical domain of the output variables adjusted by the PID control parameters of $\triangle k_p \triangle k_d$ is contained in [−1, 1]. The fuzzy subsets are described in fuzzy language as positive big (PB), positive medium (PM),

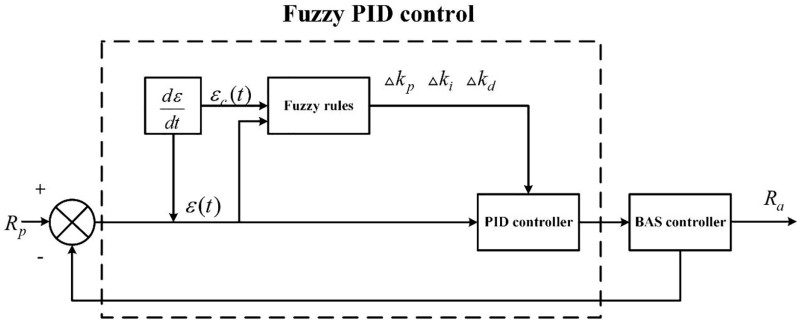

**Figure 5 The system structure of the fuzzy PID controller.**

positive small (PS), zero (ZO), negative small (NS), negative medium (NM), and negative big (NB) to facilitate data processing. Each fuzzy subset uses the triangular affiliation function corresponding to each fuzzy subset.

To ensure the reliability and stability of the whole system, reasonable fuzzy rules are proposed as follows:

(1) When $\varepsilon$ and $\varepsilon_c$ are small or equal to 0, $k_p$ takes a large value, $k_i$ takes a large value, and $k_d$ takes a medium value.

(2) When $\varepsilon$ and $\varepsilon_c$ are large, $k_p$ takes a large value, $k_i$ takes a zero value, and $k_d$ takes a small value.

(3) When $\varepsilon$ and $\varepsilon_c$ are equal in size, $k_p$ takes a small value, $k_i$ takes a moderate value, and $k_d$ takes a moderate value.

According to the above fuzzy rules, the fuzzy control rules are established as shown in Table 1.

## The improved seeker optimization algorithm (ISOA) optimized fuzzy PID controller

SOA is a new swarm intelligent evolutionary algorithm that can find the optimal solution to a target by simulating the human search behavior for both the position and direction in space (*Xiangrui et al., 2022*). In the conventional fuzzy PID control, the quantization and scaling factors must be set according to the expert's experience and cannot be adjusted adaptively when they are set. Thus, poor settings of these parameters could lead to oscillation and overshoot of the system, which cannot adapt to the random errors and disturbances existing in the actual regulation and control, and the steady-state performance becomes poorer. To improve the performance of the fuzzy PID controller, the SOA is employed for both the input and output processes of the fuzzy controller.

The SOA provides a basic principle. Suppose that a population has $S$ search individuals distributed in a $D$-dimensional space. Everyone is equipped with both position and direction information. Through multiple search updates, the individual obtains both the optimal position and direction in the space to attain the objective function's optimal solution. The position of individual $X_i$ is shown in Eq. (3) (*Duan, Luo & Liu, 2022*).

**Table 1 Fuzzy control rule table.**

| $\Delta k_p, \Delta k_i, \Delta k_d$ | | $\varepsilon$ | | | |
|---|---|---|---|---|---|
| | | *NB* | *NM* | *NS* | *ZD* |
| $\varepsilon_c$ | NB | PB/NB/PS | PB/NB/NS | PM/NM/NB | PM/NM/NB |
| | NM | PB/NB/PS | PB/NB/NS | PM/NM/NB | PS/NS/NM |
| | NS | PM/NB/ZO | PM/NM/NS | PS/NM/NM | PS/NS/NM |
| | ZO | PM/NB/ZO | PM/NM/NS | PS/NM/NM | PS/NS/NM |
| | PS | PS/NM/ZO | PS/NS/ZO | PS/PM/PM | NS/PS/PS |
| | PM | PS/ZO/PB | ZO/ZO/NM | NS/PS/PS | NM/PS/PS |
| | PB | ZO/ZO/PB | ZO/ZO/PM | NM/PS/PM | NM/PM/PM |

| $\Delta k_p, \Delta k_i, \Delta k_d$ | | $\varepsilon$ | | |
|---|---|---|---|---|
| | | *PS* | *PM* | *P* |
| $\varepsilon_c$ | NB | PS/NS/NB | ZO/ZO/NM | ZO/ZO/PS |
| | NM | PS/NS/NM | ZO/ZO/NM | ZO/NS/PS |
| | NS | PS/NM/NM | NS/PS/PS | NS/PS/ZO |
| | ZO | NS/PS/NS | NM/PM/NS | NM/PM/ZO |
| | PS | NS/PM/PM | NM/PM/ZO | NM/PB/ZO |
| | PM | NM/PM/PS | NM/PB/PS | NB/PB/PB |
| | PB | NM/PM/PS | NB/PB/PS | NB/PB/PB |

$$X_i = [x_i, x_{i2}, \cdots, x_{in}]^p \tag{3}$$

Each individual in the group possesses three search behaviors, which are co-altruistic behavior $\vec{d}_{i,\text{coa}}$, altruistic behavior $\vec{d}_{i,alt}$, and proactive behavior $\vec{d}_{i,pro}$.

$$\begin{cases} \vec{d}_{i,\text{coa}}(t) = \vec{P}_{i,\text{best}} - \vec{x}_i(t) \\ \vec{d}_{i,\text{alt}}(t) = \vec{g}_{i,\text{best}} - \vec{x}_i(t) \\ \vec{d}_{i,\text{pro}}(t) = \vec{x}_i(t_1) - \vec{x}_i(t_2) \end{cases}. \tag{4}$$

where $\vec{x}_i(t)$, $\vec{P}_{i,\text{best}}$, and $\vec{g}_{i,\text{best}}$ represents individual current best position, individual historical best position, and group best position in the neighborhood, respectively, and $\vec{x}_i(t_1)$, $\vec{x}_i(t_2)$ are delineated as the current moment as the previous reference moment and best position of the two moments before the current moment are defined.

Once the three search behaviors of an individual are calculated, the position can be updated iteratively according to the individual's final search direction. The iterative update of the position according to the final search direction of the individual is presented in Eq. (5) (*Ning et al., 2021*).

$$\begin{cases} \vec{d}_{id}(t) = sign\left[ \omega \vec{d}_{id,\text{pro}} + 0.5\vec{d}_{id,\text{ego}} + 0.5\vec{d}_{id,\text{alt}}(t) \right] \\ \Delta x_{id}(t+1) = a_{id}d_{id}(t) \\ x_{id}(t+1) = x_{id}(t) + \Delta x_{id}(t+1) \end{cases} \tag{5}$$

where $\omega$ representing inertia weight determines the direction of the individual's self-interested movement at the next moment, $\vec{d}_{id}(t)$ shows the direction of the individual's

search at the next moment, $a_{id}$ represents the search step determined by the Gaussian affiliation function, and $x_{id}(t+1)$ denotes the location of the individual at the next moment.

To evaluate the merit of the solution in the search evolution process, the ITAE performance index is chosen as the minimum objective function, where the squared input term $u^2(t)$ is added to reduce the overshoot and oscillation of the system, and the specific fitness function is presented in Eq. (6).

$$F = \int_0^\infty (\omega_1|e(t)|) + \omega_2 u^2(t)dt \tag{6}$$

where $\omega_1$ and $\omega_2$ take 0.9 and 0.1, respectively.

In the SOA, inertia weights play an important role in the search direction of individuals. When the individual target values are not attained, the individual search ability becomes poor, and the inertia weight should be increased to improve the global search ability. When the individual search values are scattered, the weight should be reduced to speed up the convergence. However, the inertia weight ω is generally fixed in the standard SOA, and this strategy will not be able to cope with the random disturbance factors of the fuzzy control process. Therefore, this article improves the SOA by utilizing a weight factor with adaptive characteristics to search for the best. Equation (7) presents this.

$$\omega = \begin{cases} \omega_{\min} + \dfrac{(\omega_{\max} - \omega_{\min}) \cdot (f - f_{\min})}{f_{\text{avg}} - f_{\min}}, & f \leq f_{\text{avg}} \\ \omega_{\max} \end{cases} \tag{7}$$

where $\omega_{\max}$ and $\omega_{\min}$ show the maximum and minimum values of the inertia weights, respectively, $f$, $f_{\min}$, and $f_{\text{avg}}$ designates the current target fitness value, the minimum value of fitness in the current group of individuals, and the mean value of fitness in the current group of individuals, respectively.

When the calculated fitness score is lower than the population mean, the adaptive weight $\omega$ is employed for iteration; otherwise $\omega_{\max}$ is chosen as the individual iteration weights that can be adaptively adjusted according to the changes in the population fitness values. More up-to-date research is also available (*Liu et al., 2023*; *Manuel, İnanç & Lüy, 2023*; *Zhang & Xiao, 2023*).

## Design of the ISOA-Fuzzy PID Controller

The structure of the ISOA-fuzzy PID control can be obtained from "The ISOA-optimized fuzzy PID controller", as shown in Fig. 6.

Based on the above study, a BAS controller with the SOA-optimized fuzzy PID is designed, and the specific steps are presented as follows:

Step 1: Initialize the search population, set the SOA individual size $N$ as 50, set the individual search space dimension $D$ as 5, represent the quantization factor and scale factor of fuzzy control, set the individual search range as (0; *Basnayake et al., 2017*), set the maximum number of iterations $M$ as 50, set the minimum fitness value $f$ as 0.1, and set the inertia weights $\omega_{\max}$ and $\omega_{\min}$ as 0.9 and 0.1, respectively.

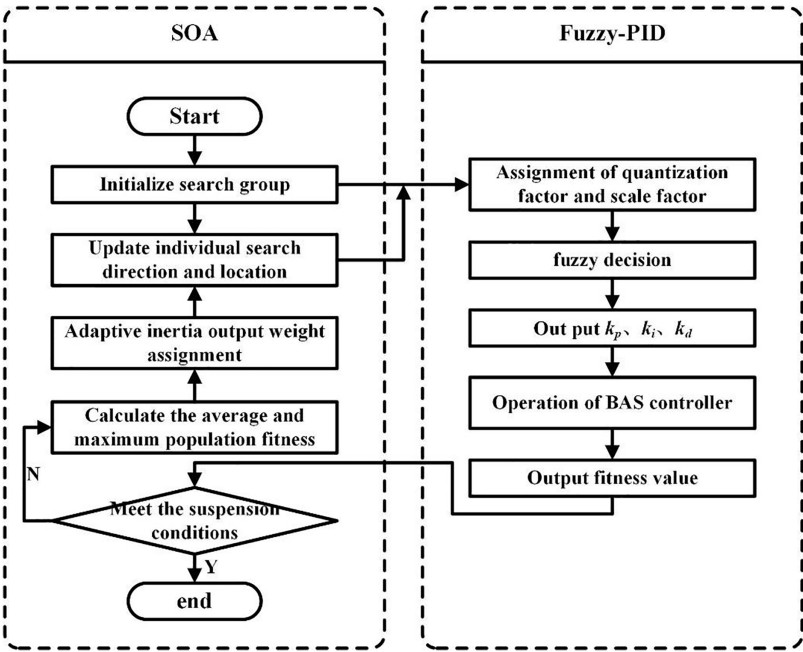

**Fuzzy PID control**

**Figure 6 The structure of ISOA-fuzzy PID control-based.**

**Figure 7 The coordinated control flow of BAS with ISOA-optimized fuzzy PID.**

Step 2: Call the BAS fuzzy PID module to obtain the fitness value of each individual in the current iteration and determine whether the abort condition (the maximum number of iterations or the minimum fitness value) is satisfied.

Step 3: Calculate the maximum and mean values of the population fitness in the current iteration and assign adaptive inertia weights ω to each of the N individuals according to Eq. (4).

Step 4: Update both the orientation and position of individuals according to Eq. (5) and go back to Step 2.

The coordinated control flow of the BAS with the SOA-optimized fuzzy PID is shown in Fig. 7.

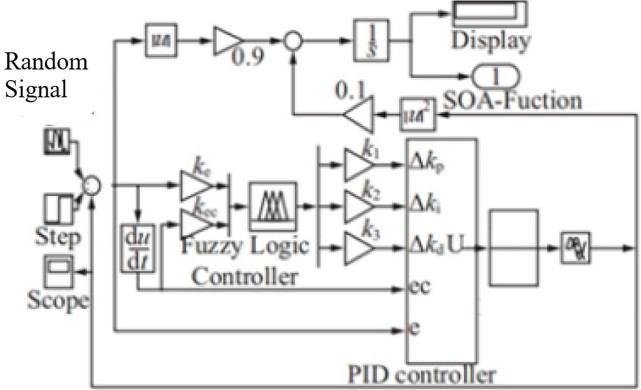

**Figure 8 The BAS coordinated control module.**

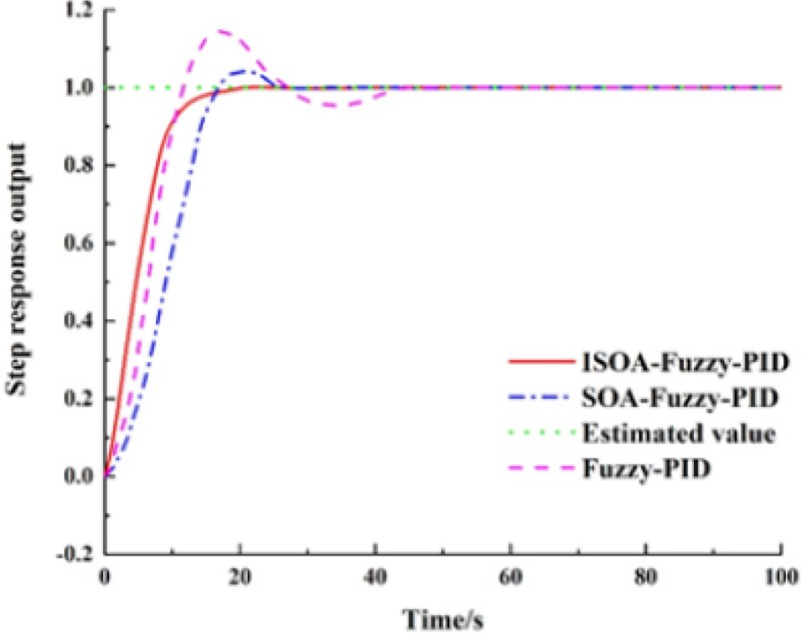

**Figure 9 The system response curves for the fuzzy PID, SOA-PID, and ISOA-PID control methods.**

## SOLUTION TESTING

To analyze the performance of the ISOA-optimized fuzzy PID control model in the BAS coordinated control framework, the Simulink module in MATLAB 2017 is conducted to run simulation and comparison experiments, which is shown in Fig. 8.

This model sets the system input to a first-order step response. Running simulations lead the ISCS to send the command to the preset -value. The system response curves for the Fuzzy-PID, SOA-Fuzzy-PID, and ISOA-Fuzzy-PID control methods are shown in Fig. 9.

According to the response curve in Fig. 10, the Fuzzy-PID control is applied to BAS. When the coordinated control is applied to BAS, a faster rise process is observed, and the

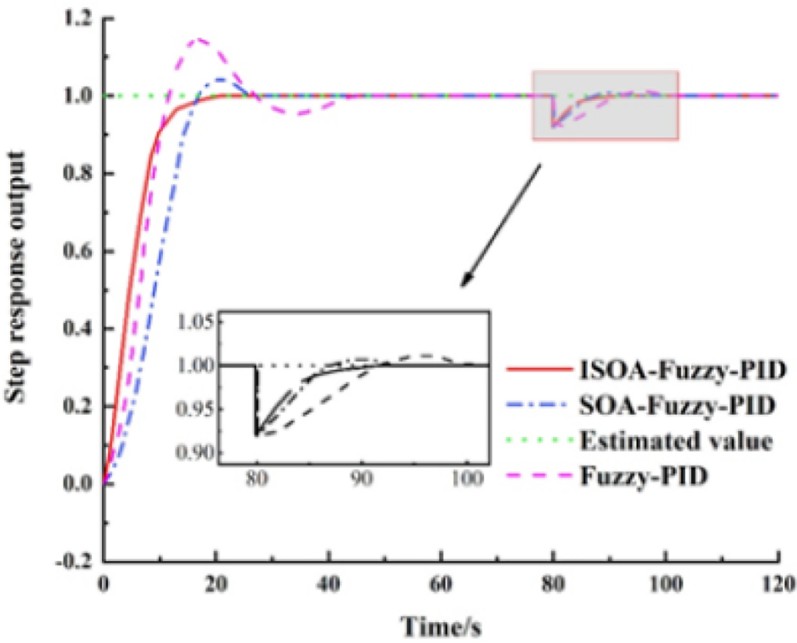

**Figure 10 Step response curve of the system with interference.**

**Table 2 Contrast of various performance indicators of the control system.**

| Method | Overshoot/% | Peak time/s | Adjustment time/s | Steady-state error/% |
|---|---|---|---|---|
| Fuzzy-PID | 14.6 | 16.9 | 47.1 | 0.8 |
| SOA-Fuzzy-PID | 4.1 | 20.3 | 26.4 | 0.4 |
| ISOA-Fuzzy-PID | 0 | 19.3 | 19.1 | 0.01 |

preset value is reached first. However, due to the nonlinearity of the controlled object, the Fuzzy-PID control is prone to overshoot, and the final overshoot is 14.6%, the regulation time is 47.1 s, and the steady-state error is 0.8%, so the system performance and the steady-state performance are poor. The overshoot of the SOA-Fuzzy-PID model is 4.1% and the adjustment time is 26.4 s. Although this model can effectively suppress the overshoot, a slow rise process appears and the parameters in the controller still need to be adjusted empirically, so the adaptive capability is not good. The ISOA-Fuzzy-PID model has no overshoot in the control process, and when compared with the SOA-Fuzzy-PID, the overshoot is reduced by 4.1% and the regulation time is shortened by 27.8%, so its control performance is the best. The specific performance indexes of the three control methods are shown in Table 2.

To verify the stability and adaptive capability of the ISOA-Fuzzy-PID model when applied to the BAS coordinated control, disturbance signals, and Gaussian random signals are introduced to simulate the response after the change of real environment parameters. Figure 11 shows the step response curves of the three control systems with the introduction

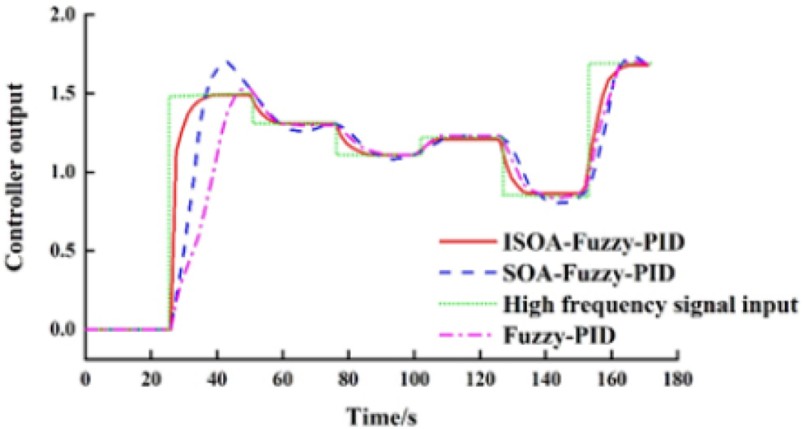

**Figure 11 The step response curve of the system with a high-frequency signal.**

of disturbances at 80 s after the output stabilization. Figure 9 depicts that after introducing an 8% disturbance of the preset value, the ISOA-Fuzzy-PID controller returns to the steady state after 10 s, while the SOA-Fuzzy-PID and the Fuzzy-PID return to the constant state after 21 and 26 s, respectively.

Figure 11 shows the output curves of the three control methods after introducing the Gaussian random signal, where the mean and the variance of the signal are set to 1 and 0.1, and the sampling time is 25 s, respectively. Figure 11 shows that the ISOA-fuzzy PID controller has better regulation ability, curve tracking, and steady-state performance than the fuzzy PID and the SOA-PID after introducing a Gaussian random signal.

In summary, the ISOA-PID control model adopted in this article can effectively realize accurate and fast responses employing the BAS coordinated control, contributing to the safe and stable operation under the ISCS framework.

## CONCLUSION

In this article, the metro BAS control method employing the ISOA-optimized fuzzy PID is adopted based on the excellent merit-seeking capability of the SOA algorithm after the SOA is improved. The experiments show that the optimization of the quantization and scaling factors of the fuzzy controller by utilizing the intelligent optimization method can improve the performance of the system control while avoiding over-reliance on the expert experience. According to the performance indexes of the simulation results, the ISOA-Fuzzy-PID control reduces the regulation time and overshoot when compared with Fuzzy-PID in the BAS coordinated control processes. The system can be adjusted to the steady state first after introducing disturbances and high-frequency signals, suggesting that the model has high control accuracy and steady-state performance and provides a reference value for the current metro BAS design.

The model is proven to have increased control accuracy and steady-state performance, which provides some reference value for the current subway BAS design. In addition, it is worth noting that the actual ISCS modeling of the metro needs to consider multiple environmental factors, and disaster data still needs to be further collected. However, the

experimental tests are not conducted in this article due to not collecting enough data for the research. In the next phase of the research, we plan to practically test the performance of the proposed method for the application of coordinated control of the BAS in the ISCS in the subway.

### Funding
This work was supported by the Key Research and Development Plan of Zhejiang Province (No. 2021C01031) and the Public Welfare Science and Technology Plan of Ningbo City (No. 2022S125). The funders had no role in study design, data collection and analysis, decision to publish, or preparation of the manuscript.

### Grant Disclosures
The following grant information was disclosed by the authors:
Key Research and Development Plan of Zhejiang Province: 2021C01031.
Public Welfare Science and Technology Plan of Ningbo City: 2022S125.

### Competing Interests
The authors declare that they have no competing interests. Hui Fang, Shusong Yang, Yang Wang, and Yue Jiang are employed by Ningbo Rail Transit Group Co., Ltd, and Ying Shi is employed by ZheJiang SUPCON Technology Co., Ltd.

### Author Contributions
- Hui Fang conceived and designed the experiments, performed the computation work, prepared figures and/or tables, authored or reviewed drafts of the article, and approved the final draft.
- Shusong Yang conceived and designed the experiments, performed the experiments, analyzed the data, performed the computation work, prepared figures and/or tables, authored or reviewed drafts of the article, and approved the final draft.
- Ying Shi conceived and designed the experiments, performed the experiments, analyzed the data, prepared figures and/or tables, authored or reviewed drafts of the article, and approved the final draft.
- Yang Wang conceived and designed the experiments, performed the experiments, performed the computation work, authored or reviewed drafts of the article, and approved the final draft.
- Yue Jiang performed the experiments, analyzed the data, performed the computation work, authored or reviewed drafts of the article, and approved the final draft.
- Chaochao Song analyzed the data, performed the computation work, prepared figures and/or tables, authored or reviewed drafts of the article, and approved the final draft.
- Wei Zhang performed the experiments, analyzed the data, authored or reviewed drafts of the article, and approved the final draft.

## Data Availability

The code and data set are available in the Supplemental Files.

## Supplemental Information

Supplemental information for this article can be found online at http://dx.doi.org/10.7717/peerj-cs.1765#supplemental-information.

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
