# Peer review of "A new cooperative control solution of subway BAS: an improved fuzzy PID control algorithm"

_PeerJ Computer Science, doi:10.7717/peerj-cs.1765_

## Round 0.1 · original submission · Major Revisions

Dear authors

Your paper has been reviewed by experts in the field and you will see that they have a couple of suggestions for improvement of your article.

I do agree with them

Please also improve the quality of English language as well as the abstract.

**Language Note:** The Academic Editor has identified that the English language must be improved. PeerJ can provide language editing services - please contact us at copyediting@peerj.com for pricing (be sure to provide your manuscript number and title). Alternatively, you should make your own arrangements to improve the language quality and provide details in your response letter. – PeerJ Staff

Reviewer 1 ·

Basic reporting

The manuscript contributes to the existing literature. However, the paper contains some severe issues that need a comprehensive revision to better present the findings of the conducted research. The list below indicates the issues as follows:
1. The title of the article should underline the contribution of the research.
2. The abstract should state the research problem, research motivation, the methodology, the contribution, the data used to implement the proposed method, and finally some key findings.
3. The article consists of long paragraphs. Shorter paragraphs based on the content should be constructed so, readers can easily follow the conducted research and other materials presented in the article.
4. All figures and table titles should be centered.
5. proofreading is a must.
6. Instead of using “equation (1)”, the abbreviated form, Eq.(1), should be used.

Experimental design

7. The authors should explain how the fuzzy rules in Table 1 are generated. Are they generated based on empirical data or based on all combinations?
8. In the article, the authors claimed that “ takes 0.9 and takes 0.1.” Why? Please discuss it and provide more remarks.
9. The titles of sections and subsections should be checked and corrected where necessary.
10. All figures and tables should be cited where they are stated. Please check all.

Validity of the findings

11. More up-to-date references should be added to the references section and discussed where necessary.
12. The conclusion section should be improved, and the future research agenda should be stated in a separate paragraph.
13. Which software is used? Did the authors run any data preprocessing steps? Please discuss it.
14. Is it possible to use other heuristic optimization algorithms to substitute the optimization algorithm used in the article? Please discuss.

Reviewer 2 ·

Basic reporting

The article has presentation and language issues. So, authors should carefully check and fix them. On the other hand, the issues related to the proposed methodology should be improved. We expect the authors to present sound responses to each issue below. A major revision is required for the article.
1. Why did the authors choose the ISOA optimization scheme? How did they decide it? Please discuss it. What are the advantages of the ISOA optimization scheme over the other ones if exists?
2. What types of membership functions are used to generate fuzzy rules? Please discuss it.
3. This sentence is taken from the text: “triangular affiliation function” What is it? Is it a triangular fuzzy number?

Experimental design

4. How did the authors determine the fuzzy rules? Please discuss it.
5. What software is used to generate simulation data?
6. Did the authors check whether the simulation data has outlier and influential points? Please discuss it.
7. Did the authors conduct and normalization process before running the proposed method?
8. Are the fuzzy rules presented in Table 1 enough to represent the whole system? Please discuss it. Is there any redundant rule that appears in the implementation process? How are the weights of the rules determined? Please discuss it.

Validity of the findings

10. All abbreviations used in the article should be checked. All abbreviations should accompany with full group of words.
11. Some numerical numbers are assumed directly in the article. Why were they chosen as they are? Please discuss and provide more remarks.
12. The proposed method should be presented in an algorithm.

---

## Round 0.2 · accepted · Accept

Dear authors

Thanks for your resubmission. Based on the input from reviewers, I am pleased to inform you that your manuscript is scientifically suitable for publication, congratulations and thank you for your fine contribution

Reviewer 2 ·

Basic reporting

All the aforementioned changes have been implemented

Experimental design

The authors have made necessary revisions.

Validity of the findings

The authors have made necessary revisions.

Additional comments

NA

Reviewer 3 ·

Basic reporting

Author has done required modifications ; You can now accept the paper

Experimental design

no comment

Validity of the findings

no comment